# Polymorphisms of the Growth Hormone Releasing Hormone Receptor Gene Affect Body Conformation Traits in Chinese Dabieshan Cattle

**DOI:** 10.3390/ani12131601

**Published:** 2022-06-21

**Authors:** Shuanping Zhao, Hai Jin, Lei Xu, Yutang Jia

**Affiliations:** Anhui Province Key Laboratory of Livestock and Poultry Product Safety Engineering, Institute of Animal Husbandry and Veterinary Medicine, Anhui Academy of Agricultural Sciences, Hefei 230031, China; zhaoshuanping@163.com (S.Z.); jinhaizjm@163.com (H.J.); yutang2018@163.com (Y.J.)

**Keywords:** growth hormone-releasing hormone receptor gene, Chinese Dabieshan cattle, body conformation traits, association analysis

## Abstract

**Simple Summary:**

Increasing growth traits can impact the economic benefit of livestock. Polymorphisms can offer us potential molecular markers for the growth traits of cattle, and provide a reference for cultivating an improved cattle population according to the markers in the future. In this study, the growth hormone-releasing hormone receptor gene polymorphism is significantly associated with the body conformation traits of Chinese Dabieshan cattle, and it could be used as a candidate genetic marker for body conformation trait selection in cattle. This study further verified that the *GHRHR* gene plays a vital role in Chinese cattle, and provides a data basis for the research and improvement of cattle breeding programs in China.

**Abstract:**

This study was performed to expose the polymorphisms of the growth hormone-releasing hormone receptor gene in Chinese Dabieshan cattle, evaluate its effect on body conformation traits, and find potential molecular markers in Chinese cattle. The *GHRHR* structure and the phylogenetic tree were analyzed using bioinformatics software. The polymorphism of the *GHRHR* gene in 486 female cattle was genotyped by PCR-RFLP and DNA sequencing, and the association between SNPs and body conformation traits of Chinese Dabieshan cattle was analyzed by one-way ANOVA in SPSS software. *GHRHR* was often conserved in nine species, and its sequence of cattle was closest to sheep and goats. Six polymorphic SNPs were identified, g.10667A > C and g.10670A > C were missense mutation. The association analysis indicated that the six SNPs significantly influenced the body conformation traits of Chinese Dabieshan cattle (*p* < 0.05). Six haplotypes were identified and Hap1 (-CAACGA-) had the highest frequency (36.10%). The Hap3/5 (-GCCCCCGGAAGG-) exhibited a significantly greater wither height (WH), hip height (HH), heart girth (HG), and hip width (HW) (*p* < 0.05). Overall, the polymorphisms of *GHRHR* affected the body conformation traits of Chinese Dabieshan cattle, and the *GHRHR* gene could be used as a molecular marker in Dabieshan cattle breeding programs.

## 1. Introduction

Dabieshan cattle area Chinese native breed and are widely rearedin the middle and lower reaches of the Yangtze River [1]. Based on the mitochondrial DNA analysis and Y-SNPs and Y-STRs markers tests, Dabieshan cattle belong to the Southern cattle type in China, have two types of maternal origin from *Bos taurus* and *Bos indicus*, and the displacement loop regions had rich genetic diversity (π = 0.023) [2,3]. Compared with exotic commercial cattle breeds, Dabieshan cattle have some important traits, including adaptation to low-quality feed resources, fine fat deposition capabilities, wet–heat tolerance and disease resistance, whereas slow growth was prioritized for genetic improvement. The molecular marker-assisted selection provided a new strategy to improve the production traits of livestock. Selecting the potential loci and understanding the association between gene variation and body conformation traits are useful for accelerating the improvement of Dabieshan cattle.

The growth hormone-releasing hormone receptor (*GHRHR*) is a member of the superfamily of the G protein-coupled receptors subfamily [4]. It works together with the growth hormone-releasing hormone (*GHRH*) to regulate the growth hormone (*GH*) axis and the development and proliferation of pituitary somatotropes. In addition, the *GH* regulates pituitary *GHRHR* expression in rodents [5]. In humans, a number of different mutations including promoter mutations [6], nonsense mutations [7], splicing mutations [8], missense mutations [9], and silent mutations [10] were identified in the *GHRHR* and resulted in isolated growth hormone deficiency (IGHD). Godfrey et al. also indicated that a missense mutation in mouse *GHRHR* has reduced *GH* secretion and resulted in a dwarf phenotype [11]. In Lanzhou fat-tail sheep population, a *GHRHR* deletion mutation caused significant body weight loss [12], and the mutations in the *GHRHR* significantly affect body length (BL) and body height (BH) of the Guanzhong goat and Xinong Sannen dairy goats [13]. In Nanyang cattle, a mutation in 5′UTR of the *GHRHR* significantly influenced the body weight (BW)at6 months, and average daily gain (ADG) at 0–6 months and 6–12 months (*p* < 0.05) [14]. These results suggested that the *GHRHR* plays a vital role in the growth and development of livestock. 

The aims of the study were to characterize the structure of the *GHRHR*, identify its SNPs and explore the effect on body conformation traits in Dabieshan cattle. The sequence alignment, phylogenetic and motif analysis were investigated to reveal the similar structure and functions of the *GHRHR* in different species. Six novel SNPs were found by PCR-RFLP and sequencing in 486 Chinese Dabieshan cattle, the haplotypes and their association with body conformation traits were identified. These results indicated that the *GHRHR* gene could be used as a molecular marker in Dabieshan cattle breeding programs. Our study provides a data basis for the scientific conservation, exploitation and utilization of Dabieshan cattle.

## 2. Materials and Methods

### 2.1. Ethics Statement

The experimental animals and procedures performed in this study were approved by the Animal Care and Use Committee of the Anhui Academy of Agricultural Sciences (approval number A19-CS20). 

### 2.2. Sequence Alignment, Phylogenetic Analysis, Motif Analysis, and Predicted Structures

The amino acid sequences of *GHRHR* gene in nine species, including human (*Homo sapiens* NP_000814.2), rat (*Rattus norvegicus* NP_036982.1), mouse (*Mus musculus* NP_001003685.2), cattle (*Bos taurus* NP_851363.1), horse (*Equus caballus* XP_001916581.2), goat (*Capra hircus* XP_005679295.2), sheep (*Ovisaries* NP_001009454.3), pig (*Sus scrofa* NP_999200.1), and chicken (*Gallus gallus* NP_001032923.1), were acquired through NCBI database. Multiple sequence alignment in DNAMAN software was used to analyze the obtained sequence, and the phylogenetic tree was constructed using neighbor-joining phylogenetic tree in MEGA (version 7.0.26) software [15]. The motifs of *GHRHR* proteins were investigated through MEME suite (http://meme.nbcr.net/, accessed on 12 February 2022) to reveal their structure, characteristics and function [16]. The CDD NCBI (https://www.ncbi.nlm.nih.gov/Structure/cdd/cdd.shtml, accessed on 15 February 2022) was performed to analyze conserved domains [17], and the SWISS-MODEL (http://swissmodel.expasy.org/, accessed on 18 April 2022) was used to predict the 3D structures.

### 2.3. Phenotypic Data and DNA Sample Collection

A total of 486 female Dabieshan cattle aged 24 to 30 months were randomly chosen from National Species Resources Protection Farm (Anqing, China). According to NRC standards (Nutrient Requirement of Beef Cattle), the animals on the farm were fed a total mixed ration (TMR), which comprised 25% concentrate and 75% roughages of dry straw and corn silage, and water was offered ad libitum.

The phenotype performance for the body conformation traits in Dabieshan cattle, including body length (BL), wither height (WH), hip height (HH), heart girth (HG), abdominal girth (AGR), hip width (HW), and pin bone width (PBW), were collected following the methods of Wang et al. [18] and Yang et al. [19]. Ear marginal tissues collected from these animals were used to extract genomic DNA according to the TIANamp Genomic DNA Kit (TIANGEN, Beijing, China) procedure. Genomic DNA were measured with the spectrophotometer, and then diluted to 50 ng/μL for polymerase chain reaction (PCR) analysis. 

### 2.4. Primers, Polymerase Chain Reaction, and Gel Electrophoresis

Primer Premier 5 software (PREMIER Biosoft International, San Francisco, CA, USA) was used to design the primers (Table 1) for amplifying the bovine *GHRHR* gene (GenBank accession number: NC_037331.1). The fragments were amplified with 25 µL of reaction volume, which contained 1.0 μL of each forward and reverses primers (100 ng/μL), 1 µL genomic DNA (50 ng/µL), 12.5 uL of 2 × Taq Mix (TIANGEN, Beijing, China), and 9.5 µL double distilled water (ddH_2_O). The PCR amplification was performed according to the procedure, including pre-denaturation at 94 °C for 5 min, 32 cycles of denaturation at 94 °C for 30 s, annealing at 60 °C for 30 s, and extension at 72 °C for 30 s, followed by a final extension at 72 °C for 5 min. The PCR products of F1R1 were genotyped by PCR-RFLP. The digestion reaction mixture contained 5 uL PCR products, 1 uL 10 × buffer, 10 U restriction enzyme (PshAI, NEB, Ipswich, MA, USA), and sterile water. Samples were incubated at 37 °C for restriction enzyme overnight, and the digested products were analyzed by 1.5% agarose gel electrophoresis, stained with GeneGreen Nucleic Acid Dye (TIANGEN, Beijing, China) in 1% TBE. The PCR products of F2R2 were also electrophoresed on a 1.5% agarose gel, and then purified and sequenced through Sangon Biotech (Shanghai, China).

### 2.5. Statistical Analysis

The allelic and genotypic frequencies of six SNPs were measured by direct counting.Hardy–Weinberg equilibrium (HWE) was assessed using POPGENE 3.2 software package with the Chi-squared (χ^2^) test. Population genetic indexes, such as gene heterozygosity (H_e_), gene homozygosity (H_o_), effective allele numbers (N_e_) and polymorphism information content (PIC), were analyzed according to Chakraborty and Nei [20]. LD plot was performed to analyze the Linkage disequilibrium (LD)of SNPs and haplotype block of Haploview software was used to obtain the haplotypes [21,22]. The diplotype was assessed according to the SNPs and haplotypes.

The association analysis was calculated by one-way ANOVA in the SPSS software (version 24.0) (Armonk, NY, USA), which was used in our previous study [23]. The general linear model was used to determine the association between SNPs and body measurement traits. The equation was shown as follows:y_ijk_ = u + g_i_ + a_j_ + s_k_ + e_ijk_(1)where y_ijk_ was the phenotypic observation, u was the mean, g_i_ was the fixed effect of genotype, a_j_ was the random effect of sire, s_k_ was the fixed effect of age, and e_ijk_ was the residual effect. The results were presented as the mean ± standard error, and *p* < 0.05 was considered a significant difference.

## 3. Results

### 3.1. Species Homology, Phylogenetic Tree, and Conservative Estimation

The cDNA of the *GHRHR* in cattle consisted of 102-bp 5′UTR sequences, followed by a 1326-bp open reading frame and 993-bp 3′UTR. A total of 441 amino acid residues were encoded in the coding region with a molecular weight of 49.25 kD and an isoelectric point of 6.37. In Table 2, both the 5′-donor and 3′-acceptor splice sites in the *GHRHR* conformed to the GT–AG rule. Figure 1 shows the conservative analysis of the *GHRHR* protein in nine animals (human, rat, mouse, cattle, pig, goat, sheep, horse, and chicken), and signal peptide, HormR, and Pfam were often observed. The neighbor-joining phylogenetic tree (Figure 2) revealed that cattle were closest to goats and sheep for the *GHRHR* sequence, while humans, rats, mice, and chickens were fartherfrom the cattle branch. In total, 12 significant motifs were found using MEME online in the super secondary protein structure of the *GHRHR* gene (Figure 3), indicating that it plays a similar role in the nine animals.

### 3.2. Identification of SNPs in GHRHR

In this study, six SNPs including g.2052C > G in intron 1, g.10667A > C and g.10670A > C in exon 10, and g.10684C > G, g.10699A > G, and g.10763A > G in intron 10, were identified in the *GHRHR* by PCR-RFLP and sequencing. The agarose gel electrophoresis of PCR-amplified products using F1R1 and F2R2 primers were shown in Figure 4a,b, and the PCR-*PshAI*-RFLP analysis at g.2052C > G locus was present in Figure 4c. The 498 bp PCR fragments of F2R2 primers can be digested by *PshAI* into 289 bp and 209 bp for allele C. The sequenced map of five SNPs in the *GHRHR* was shown in Figure 5. All these SNPs are biallelic.

The SNPs in exon10 were missense mutations (g.10667A > C: Glutamine (Gln)→ Proline (Pro); g.10670A > C: Histidine (His)→Pro). Figure 6 shows the predicted 3D structure and the value of quality model energy analysis (QMEAN) of the normal and mutant *GHRHR* (g.10667A > C and g.10670A > C). The figures indicated there was no change in the 3D structure (Figure 6a–d), whereas the QMEAN value of the mutant-type was significantly different from the wild-type (Figure 6e–h). As shown in Figure 6e,f, the mutation of Gln to Pro at the 323 sites led to the value of QMEAN increasing from −4.70 to −4.73, however, the torsion and all atoms of the wild-type had a lower level than the mutant-type, while the solvation and Cβ were opposite. Figure 6e,g show that the conversion of His to Pro at the 324 sites leads to a decrease in the value of QMEAN, solvation, torsion, Cβ, and all atoms. Additionally, the conversion of both Gln and His to Pro caused the rise of all atoms and torsion, and reduction of QMEAN, Cβ and solvation (Figure 6e,h).

### 3.3. Genetic Diversity, and Hardy Weinberg Equilibrium

In the Dabieshan cattle, 486 female cattle were genotyped by PCR-RFLP and sequencing. The genetic diversity including genotypic, allelic frequency and genetic indexes of the six detected SNPs in the *GHRHR* was evaluated in Table 3. The G allele at g.10699A > G locus was most prevalent (0.71), whereas the GG genotype at g.10684C > G was more frequent than others. The values of heterozygosity (H_e_) were 0.414~0.499 and effective allele numbers (N_e_) values were1.707~2.000. Polymorphism information content (PIC) values ranged from 0.328 to 0.375, indicating that the genetic diversity of Dabieshan cattle at these six SNP sites was at a medium level (0.25 < PIC < 0.5).

### 3.4. Linkage Disequilibrium, and Haplotype Analysis

Chi-square tests (*p* < 0.05) were calculated for the Hardy–Weinberg equilibrium (HWE) in detected SNPs. Table 3 shows that six SNPs except forg.10763A > G were in the Hardy–Weinberg equilibrium. Table 4 presents the linkage disequilibrium among the detected SNPs, the D’ values were in the range of 0.216 and 1.000, and r^2^ values were on a scale of 0.012 to 0.919. Several studies indicated that r^2^ is more suitable for measuring LD because of its insensitivity to allele frequencies [24]. The r^2^ values in this study showed that g.10670A > C, g.10684C > G, and g.10763A > G were in a relatively strong linkage, and g.10667A > C was relatively strongly linked with g.10699A > G in Dabieshan cattle (r^2^ > 0.33).

The haplotype analysis at the six loci of the *GHRHR* gene was illustrated in Table 5. Six dominating haplotypes were identified in the Dabieshan cattle, Hap1 (-CAACGA-) was the most frequent (36.10%), followed by Hap2 (-GACGGG-), Hap3 (-CCCGAG-), and Hap4 (-GAACGA-). Hap5 (-GCCGAG-) and Hap6 (-CACGGG-) showed the lowest frequency in our analysis. 

### 3.5. Association Analysis 

Table 6 illustrates the association analysis of the polymorphic locus with body conformation traits. At the g.2052C > G site, animals with genotype CC exhibited a significantly greater WH, HH and AGR in comparison with genotype GG (*p* < 0.05). For the g.10667A > C and 10670A > C locus, the influence of the AC genotype resulted in a higher PBW (*p* < 0.05), whereas in the g.10699A > G locus, animals carrying the AA genotype had a greater PBW than genotype AG (*p* < 0.05). For g.10684C > G, the GG genotype revealed a positive effect on WH and PBW (*p* <0.05). At the g.10763A > G locus, animals with genotypes AA and GG showed greater WH than the AG genotypes (*p* < 0.05).

The association between diplotype and body conformation traits was analyzed in Table 7. In Dabieshan cattle, animals carrying the Hap3/5 (-GCCCCCGGAAGG-) exhibited a significantly greater WH, HH, HG, and HW than the others (*p* < 0.05). The frequencies of diplotypes < 5.0% were not considered.

## 4. Discussion

The *GHRHR* was reported to encode the *GHRH* receptor, and its activation would release the *GH* stored in secretory granules and produce *GH* through cAMP-dependent *GH1* transcription [25]. Previous studies found that the *GHRHR* plays a pivotal role in complex biological process, especially multiple inflammatory processes [26], cell proliferation and development [27], and body conformation traits [12,13,14]. In this study, the GT–AG rule was appropriate for the *GHRHR* in bovines, indicating that it conforms to the splice site of the eukaryotic gene. The high homology, similar structural domain and common significant motifs of the *GHRHR* protein in nine species suggest that the *GHRHR* had similar functions in the nine species, which provided guidance for further studying in Dabieshan cattle. The phylogenetic tree in this study was in accordance with Wu et al. [28] and Zhao et al. [23], which showed that cattle were closest to goats and sheep, suggesting that cattle are more closely related to the sheep and goats than other domesticated animals.

Dabieshan cattle are one of the most Chinese native cattle breeds, well known for meat quality and well-adapted to Chinese production systems. Six novel SNP sites were detected and their effect on body conformation traits of Chinese Dabieshan cattle was analyzed in our study. The observed SNPs are all biallelic. The PIC values showed that the *GHRHR* was atan intermediate level (0.25 < PIC < 0.5), and had great potential as a genetic selection marker for body conformation attributes in Chinese Dabieshan cattle. The χ^2^ analysis showed that g.10763A > G was in Hardy–Weinberg disequilibrium (*p* > 0.05), which may be due to the population size and artificial selection. The r^2^-values, among the g.10670A > C, g.10684C > G, and g.10763A > G sites, and between g.10667A > C and g.10699A > G sites, were sufficiently strong for mapping. The results may be caused by the lower recombination and higher genotypic variation at these sites [29].

There were many agreeing reports regarding the effect of the *GHRHR* mutations on the body conformation traits of cattle in previous studies. Zhang et al. [14] indicated that a mutation in 5′UTR of the *GHRHR* had a significant effect on the Nanyang cattle phenotype. In this study, there was a significant association between the six identified SNPs and body conformation traits in Dabieshan cattle. Moreover, Hap3/5 (-GCCCCCGGAAGG-) had a greater WH, HH, HG, and HW, the Hap3/3 (-CCCCCCGGAAGG-) and Hap2/5 (-GGACCCGGAGGG-) exhibited greater HG and HW than animals with other diplotypes, which indicated that Hap3 (-CCCGAG-) and Hap5 (-GCCGAG-) might result in an increasing body conformation trait in Chinese Dabieshan cattle. 

Interestingly, the SNPs g.10667A > C and g.10670A > C in exon 10 were missense mutations (Gln → Pro; His → Pro). Figure 6 indicates that the QMEAN value, all atoms, and torsion of mutant 1 were higher than the wild-type and other mutants, indicating that the protein of mutant 1 might have a stronger potential energy and torsion angle in connection with pairwise atom distances than other types. The Cβ and solvation of mutant 3 were lower than those of other types, which suggest that the mutant 3 protein might exhibit a lower residue embedding ability and a weaker C-β potential energy of interaction. In accordance with the study of Wang et al. [30], the SNPs located in intron 1 and 10, show that the non-coding region variation in the *GHRHR* significantly affects the growth and development of Dabieshan cattle. Increasingly, the literature has shown that intron mutation might affect the correct processivity of the mRNA by disrupting the splice site, altering the secondary structure, affecting protein function to regulate the gene expression level or determine tissue-specific expression patterns [31].

In general, six SNPs including g.2052C > G, g.10667A > C, g.10670A > C, g.10684C > G, g.10699A > G, and g.10763A > G of the *GHRHR* gene were genotyped in Chinese Dabieshan cattle. The six detected SNPs played a significant role in the body conformation traits, and animals with Hap3/5 exhibited significantly improving body conformation traits, meaning that the *GHRHR* could be used as a marker in Dabieshan cattle breeding programs.

## 5. Conclusions

This study investigated six novel SNPs in the *GHRHR* gene in Chinese Dabieshan cattle. Our results show that all six loci contributed significantly to the body conformation traits of cattle (*p* < 0.05), and Hap3/5 was attributed to desirable body characteristics. These findings indicated that the *GHRHR* could be used as a DNA marker for improving the performance of Dabieshan cattle.

## Figures and Tables

**Figure 1 animals-12-01601-f001:**
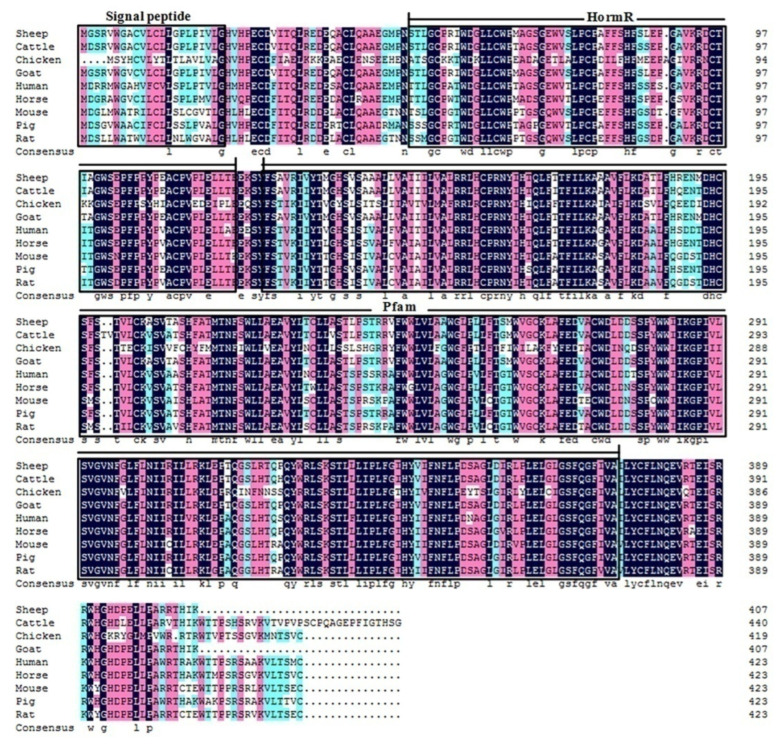
The conservative analysis of *GHRHR* protein in nine animals (Human, *Homo sapiens* NP_000814.2; Rat, *Rattus norvegicus* NP_062024.1; Cattle, *Bos taurus* NP_851363.1; Goat, *Capra hircus* XP_005679295.2; Sheep, *Ovis aries* NP_001009454.3; Horse, *Equus caballus* XP_001916581.2; Mouse, *Mus musculus* NP_001003685.2; Pig, *Sus scrofa* NP_999200.1; Chicken, *Gallus gallus* NP_001032923.1). Signal peptide was located at 1–22 sites; HormR (51–121) was present in hormone receptors, and involved in regulating G protein-coupled receptor activity; Pfam (126–374) participated in G protein-coupled receptor signaling pathways and regulated G protein-coupled receptor activity.

**Figure 2 animals-12-01601-f002:**
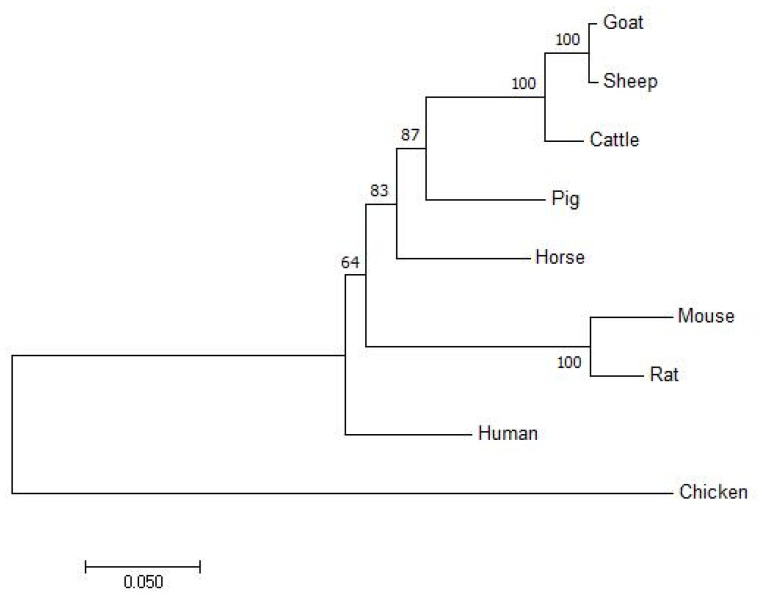
The neighbour-joining phylogenetic tree of *GHRHR* protein sequences in nine animals.

**Figure 3 animals-12-01601-f003:**
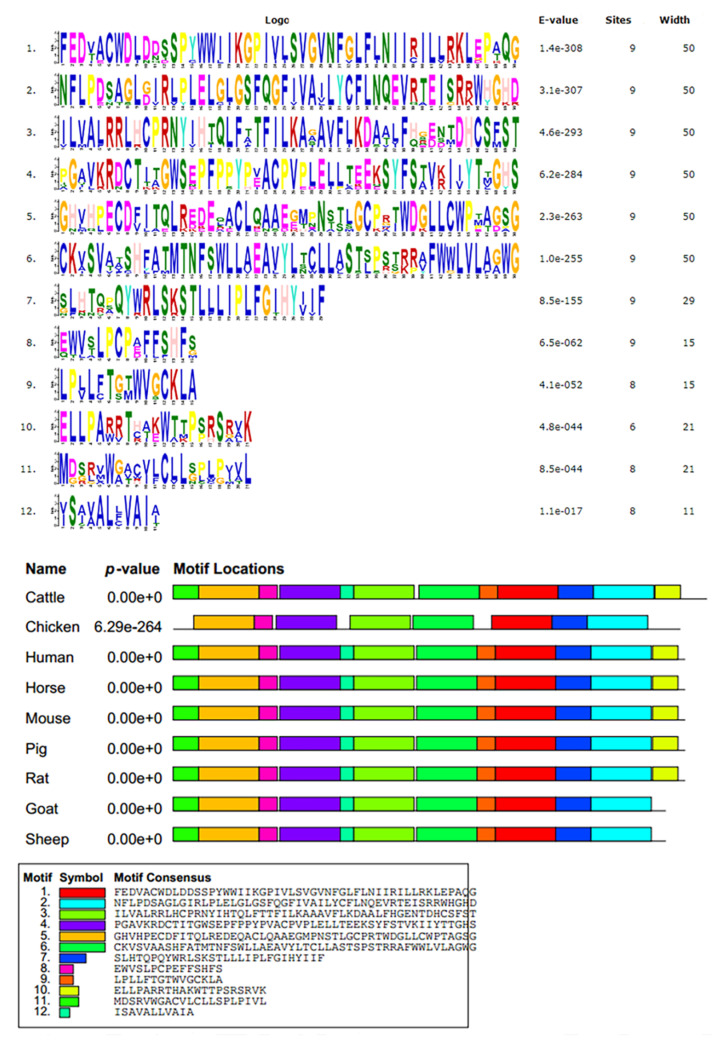
The significant motifs of *GHRHR* protein sequences in nine animals. These colors were given by MEME suit system during analysis.

**Figure 4 animals-12-01601-f004:**
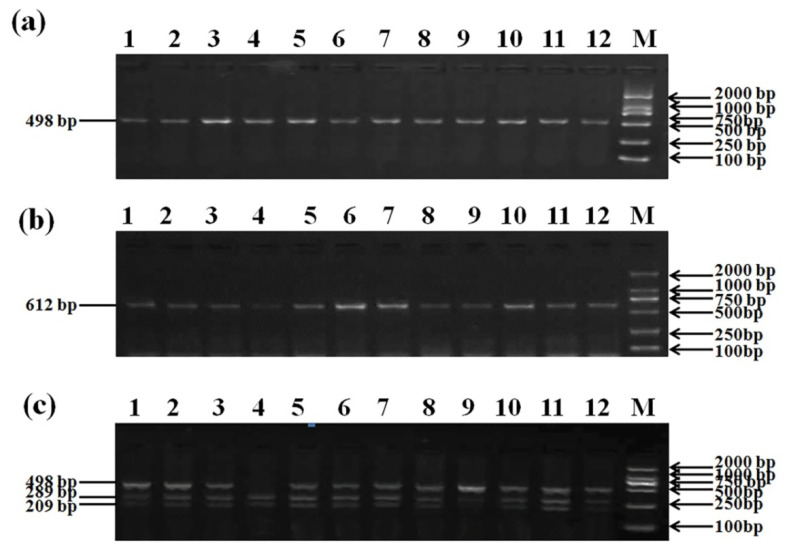
Agarose gel electrophoresis (1.5%) of PCR-amplified products in *GHRHR* gene. (**a**) F1R1 primer, (**b**) F2R2 primer, and (**c**) PCR-*PshAI*-RFLP result of SNP1 at g.2052C > G, genotype GG (498 bp), genotype GC (498 + 289 + 209 bp), and genotype CC (209 bp). Lanes 1 to 12 show the products and Lane M presents 2000 bp DNA ladder.

**Figure 5 animals-12-01601-f005:**
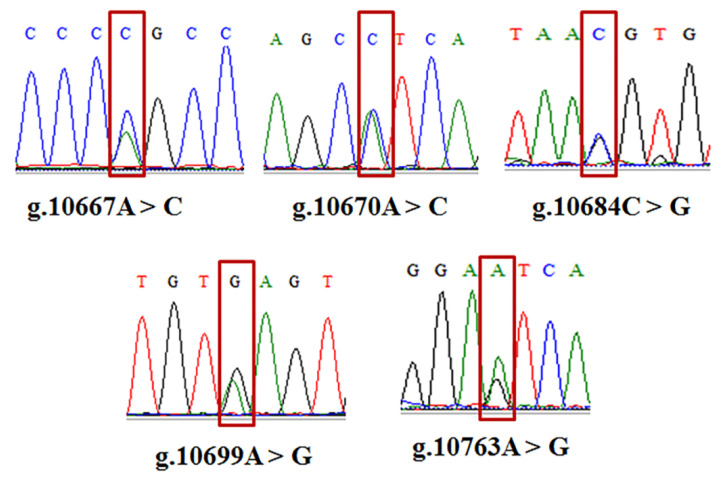
The five mutation sites with F2R2 primers in *GHRHR* gene.

**Figure 6 animals-12-01601-f006:**
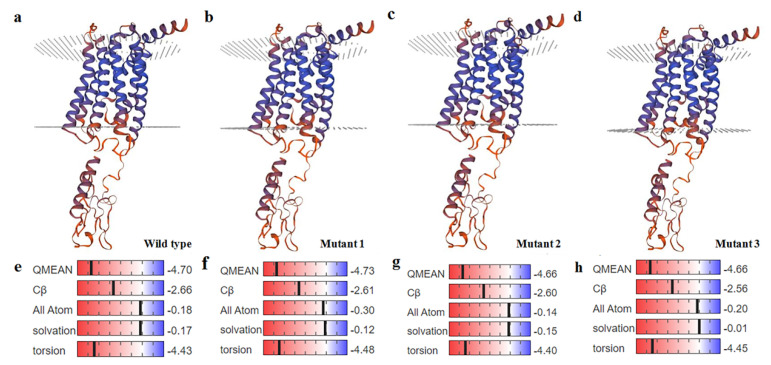
The predicted 3D structureand QMEAN of the bovine *GHRHR* gene. (**a**,**e**), wild-type; (**b**,**f**), the mutant 1-type of Gln to Pro; (**c**,**g**), the mutant 2-type of His to Pro; and (**d**,**h**), the mutant 3-type of both Gln and His to Pro.

**Table 1 animals-12-01601-t001:** Primers used for PCRanalysis in *GHRHR*.

Primer Name	Primer Sequence	Tm (°C)	Position	Segment Length
*GHRHR*-F1	5′–GGAAAGGATGGGATTGTAGT–3′	60	Intron 1	498 bp
*GHRHR*-R1	5′–CGGCAAAAGCATAGTGGGTA–3′			
*GHRHR*-F2	5′–TCCTGTCCCTCAAAGACCTG–3′	60	Intron 9, exon 10, intron 10	612 bp
*GHRHR*-R2	5′–CCCTCTCTCAAATGCTCTGG–3′			

*GHRHR*-F for forward and *GHRHR*-R for reverse.

**Table 2 animals-12-01601-t002:** The 5′-donor and 3′-acceptor splice sites in *GHRHR*.

Gene	Number	Exon Size(bp)	Intron Size (bp)	5′Splice Donor	3′Splice Acceptor
*GHRHR*	1	189	3892	CGATCgtgag	cacagGTCCT
	2	103	128	CTTGGgtacg	tctagGCTGC
	3	108	746	GCCAGgtgag	cctagGGGCT
	4	98	1150	AGGAGgtgag	cccagAAATC
	5	98	512	CTCAGgtttg	tccagGAGGC
	6	133	1410	CCACTgtaac	cacagGTCCT
	7	154	300	CTGGGgtgag	cacagGGCTT
	8	61	569	GTTGCgtgag	ttcagGTGCT
	9	70	865	TTGGGgtcag	tccagGTGAA
	10	92	456	TACTGgtaac	tgcagGCGTC
	11	130	713	TCCAGgtgag	cacagGGCTT
	12	42	2852	AAGAGgtatg	tctagGTGAC
	13	1140	–	–	–

**Table 3 animals-12-01601-t003:** Genotyping and population genetic analysis in *GHRHR*.

SNP	Genotypic Frequency			Allelic Frequency			HWE			Diversity Parameter	
	1	2	3	A1	A2	χ^2^	*p*-Value	H_o_	H_e_	N_e_	PIC
g.2052C > G	**CC**	**GC**	**GG**	C	G						
	0.28	0.54	0.18	0.55	0.45	2.16	0.339	0.505	0.495	1.981	0.372
g.10667A > C	AA	AC	CC	A	C						
	0.48	0.40	0.12	0.68	0.32	1.38	0.502	0.567	0.433	1.764	0.339
g.10670A > C	AA	AC	CC	A	C						
	0.26	0.53	0.21	0.52	0.48	1.38	0.502	0.501	0.499	2.000	0.375
g.10684C > G	**CC**	**GC**	**GG**	C	G						
	0.24	0.16	0.60	0.40	0.60	1.10	0.576	0.519	0.491	1.929	0.366
g.10699A > G	AA	AG	GG	A	G						
	0.10	0.37	0.53	0.29	0.71	3.86	0.145	0.586	0.414	1.707	0.328
g.10763A>G	AA	AG	GG	A	G						
	0.31	0.11	0.58	0.31	0.69	9.51	0.008	0.574	0.426	1.742	0.335

PIC < 0.25 represents low polymorphism; 0.25 < PIC < 0.50 represents moderate polymorphism; and PIC > 0.50 represents high polymorphism.

**Table 4 animals-12-01601-t004:** Linkage equilibrium analysis between SNP markers in *GHRHR*.

Item	g.2052C > G	g.10667A > C	g.10670A > C	g.10684C > G	g.10699A > G	g.10763A > G
g.2052C > G		r^2^ = 0.025	r^2^ = 0.046	r^2^ = 0.033	r^2^ = 0.012	r^2^ = 0.033
g.10667A > C	D’ = 0.278		r^2^ = 0.138	r^2^ = 0.124	r^2^ = 0.729	r^2^ = 0.121
g.10670A > C	D’ = 0.349	D’ = 1.000		r^2^ = 0.836	r^2^ = 0.093	r^2^ = 0.801
g.10684C > G	D’ = 0.255	D’ = 0.870	D’ = 0.992		r^2^ = 0.097	r^2^ = 0.919
g.10699A > G	D’ = 0.216	D’ = 0.919	D’ = 0.883	D’ = 0.827		r^2^ = 0.098
g.10763A > G	D’ = 0.257	D’ = 0.849	D’ = 0.992	D’ = 0.979	D’ = 0.825	

0 ≤ D’ ≤ 1 represents different degrees of linkage; r^2^ > 0.33 indicates a strong linkage. D’ = 0.278/r^2^ = 0.025 represents the degree of linkage between g.2052C > G and g.10667A > Cmarkers.

**Table 5 animals-12-01601-t005:** Haplotypes analysis of the six SNPs in *GHRHR*.

Haplotype	g.2052C > G	g.10667A > C	g.10670A > C	g.10684C > G	g.5148A > C	g.10699A > G	Frequency
Hap1	C	A	A	C	G	A	36.10%
Hap2	G	A	C	G	G	G	13.40%
Hap3	C	C	C	G	A	G	12.30%
Hap4	G	A	A	C	G	A	7.60%
Hap5	G	C	C	G	A	G	6.50%
Hap6	C	A	C	G	G	G	6.20%

Hap1, -CAACGA-; Hap2, -GACGGG-; Hap3, -CCCGAG-; Hap4, -GAACGA-; Hap5, -GCCGAG-; Hap6, -CACGGG-. Haplotype with frequency < 5.00% was excluded in analysis.

**Table 6 animals-12-01601-t006:** Association analysis of SNPs with body conformation traits in Dabieshan cattle.

SNP	Genotype	Traits (Mean ± SE)						
		BL (cm)	WH (cm)	HH (cm)	HG (cm)	AGR (cm)	HW (cm)	PBW (cm)
g.2052C > G	CC	127.76 ± 2.04	112.59 ± 1.57 ^a^	110.58 ± 1.59 ^a^	148.02 ± 1.61	174.84 ± 1.40 ^a^	31.95 ± 0.41	17.45 ± 0.55
	CG	124.93 ± 1.34	109.57 ± 1.69 ^b^	109.81 ± 1.50 ^ab^	146.39 ± 1.13	172.90 ± 1.45 ^ab^	30.41 ± 0.54	16.56 ± 0.45
	GG	123.60 ± 1.36	108.42 ± 1.03 ^b^	107.74 ± 0.93 ^b^	145.17 ± 1.62	168.64 ± 2.02 ^b^	30.90 ± 0.44	16.83 ± 0.66
	*p*-value	0.226	0.031	0.047	0.548	0.026	0.111	0.079
g.10667A > C	AA	125.38 ± 1.67	110.36 ± 1.66	109.18 ± 1.58	150.96 ± 1.33	173.42 ± 1.74	33.38 ± 0.53	16.02 ± 0.29 ^a^
	AC	125.41 ± 1.81	110.12 ± 1.54	109.19 ± 1.48	149.90 ± 1.23	175.07 ± 1.34	33.65 ± 0.48	17.35 ± 0.29 ^b^
	CC	125.87 ± 1.08	110.80 ± 1.65	109.21 ± 0.86	152.77 ± 1.33	172.54 ± 1.85	32.96 ± 0.55	16.15 ± 0.27 ^ab^
	*p*-value	0.954	0.869	0.999	0.636	0.770	0.825	0.039
g.10670A > C	AA	124.53 ± 1.27	109.92 ± 1.39	109.16 ± 1.38	147.96 ± 1.22	171.43 ± 1.97	33.65 ± 0.63	15.41 ± 0.37 ^a^
	AC	124.86 ± 1.87	109.34 ± 1.57	109.03 ± 0.96	150.22 ± 1.57	173.65 ± 1.44	33.28 ± 0.49	17.07 ± 0.29 ^b^
	CC	125.99 ± 0.97	110.88 ± 1.60	109.23 ± 1.46	152.34 ± 0.96	174.26 ± 1.66	33.23 ± 0.49	16.31 ± 0.28 ^ab^
	*p*-value	0.540	0.252	0.976	0.159	0.486	0.861	0.010
g.10684C > G	CC	124.53 ± 1.17	107.12 ± 1.16 ^b^	109.46 ± 0.81	146.40 ± 1.74	171.98 ± 1.85	33.07 ± 0.60	15.81 ± 0.48 ^b^
	CG	124.90 ± 0.89	110.45 ± 1.74 ^a^	108.00 ± 1.52	149.96 ± 1.41	172.33 ± 1.47	33.28 ± 0.51	16.29 ± 0.28 ^ab^
	GG	126.03 ± 1.97	110.91 ± 1.62 ^a^	109.22 ± 1.46	152.48 ± 1.26	174.46 ± 1.67	34.61 ± 0.42	17.15 ± 0.35 ^a^
	*p*-value	0.492	0.012	0.501	0.083	0.480	0.415	0.105
g.10699A > G	AA	126.63 ± 1.13	110.74 ± 1.72	108.85 ± 1.58	153.65 ± 1.37	173.05 ± 1.87	32.80 ± 0.58	17.13 ± 0.28 ^a^
	AG	123.56 ± 1.76	109.51 ± 1.51	109.20 ± 1.46	148.49 ± 1.95	172.62 ± 1.40	33.77 ± 0.42	15.05 ± 0.36 ^b^
	GG	125.62 ± 1.06	110.52 ± 1.66	109.25 ± 0.85	151.16 ± 1.33	173.76 ± 1.73	33.37 ± 0.54	16.10 ± 0.29 ^ab^
	*p*-value	0.347	0.596	0.920	0.279	0.905	0.713	0.046
g.10763A > G	AA	124.31 ± 1.53	110.28 ± 1.75 ^a^	109.47 ± 1.72	150.01 ± 1.43	171.55 ± 1.78	33.13 ± 0.61	15.83 ± 0.29
	AG	124.91 ± 0.95	106.79 ± 1.51 ^b^	108.00 ± 1.50	146.91 ± 1.25	173.29 ± 1.33	34.19 ± 0.37	16.90 ± 0.36
	GG	126.13 ± 1.46	110.95 ± 1.61 ^a^	109.26 ± 0.87	152.58 ± 1.28	174.87 ± 1.66	33.29 ± 0.54	16.30 ± 0.27
	*p*-value	0.492	0.049	0.441	0.174	0.070	0.280	0.185

^a,b^ Means with different superscripts are significantly different (*p* < 0.05).

**Table 7 animals-12-01601-t007:** Associations analysis of diplotypes with body conformation traits in Dabieshan cattle.

Diplotypes	Frequency	BL (cm)	WH (cm)	HH (cm)	HG (cm)	AGR(cm)	HW (cm)	PBW (cm)
Hap 1/1	0.195	125.98 ± 1.39	110.62 ± 1.09 ^ab^	109.55 ± 0.76 ^ab^	152.57 ± 1.14 ^b^	173.65 ± 1.64	33.38 ± 0.34 ^b^	16.66 ± 0.47
Hap 6/6	0.136	123.61 ± 1.21	107.91 ± 1.13 ^a^	106.09 ± 0.99 ^a^	149.08 ± 1.70 ^ab^	171.95 ± 1.76	32.54 ± 0.70 ^ab^	16.18 ± 0.81
Hap 3/3	0.122	126.00 ± 1.10	110.03 ± 0.86 ^ab^	108.93 ± 0.84 ^ab^	154.06 ± 1.42 ^b^	173.43 ± 1.96	33.39 ± 0.62 ^b^	16.97 ± 0.44
Hap 1/4	0.084	125.71 ± 1.25	112.29 ± 0.69 ^b^	111.68 ± 0.99 ^b^	149.00 ± 1.43 ^ab^	170.71 ± 1.66	33.50 ± 0.48 ^b^	16.93 ± 0.87
Hap 2/2	0.174	125.26 ± 0.87	111.13 ± 0.92 ^b^	109.30 ± 0.93 ^ab^	153.54 ± 1.07 ^b^	174.62 ± 1.53	34.02 ± 0.48 ^b^	16.99 ± 0.58
Hap 3/5	0.054	128.61 ± 1.04	114.15 ± 0.97 ^b^	112.31 ± 0.96 ^b^	155.74 ± 1.05 ^b^	174.08 ± 1.26	34.35 ± 0.59 ^b^	16.92 ± 0.83
Hap 2/2	0.063	122.72 ± 0.98	110.17 ± 0.95 ^ab^	106.89 ± 0.81 ^a^	143.61 ± 1.14 ^a^	167.06 ± 1.40	30.89 ± 0.36 ^a^	15.88 ± 0.66
Hap 2/5	0.066	124.42 ± 0.83	110.84 ± 0.97 ^ab^	109.67 ± 0.91 ^ab^	154.23 ± 0.92 ^b^	174.95 ± 1.43	34.00 ± 0.40 ^b^	16.97 ± 0.82
*p*-value		0.889	0.036	0.012	0.027	0.977	0.042	0.057

Hap 1/1, CCAAAACCGGAA; Hap 6/6, CCAACCGGGGGG; Hap 3/3, CCCCCCGGAAGG; Hap 1/4, GCAAAACCGGAA; Hap 2/2, GGAACCGGGGGG; Hap 3/5, GCCCCCGGAAGG; Hap 2/2, GGAACCGGGGGG; Hap 2/5, GGACCCGGAGGG. ^a,b^ Means with different superscripts are significantly different (*p* < 0.05).

## Data Availability

All data is provided in the manuscript.

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
