# Peer review of "Polymorphisms of the Growth Hormone Releasing Hormone Receptor Gene Affect Body Conformation Traits in Chinese Dabieshan Cattle"

_animals, 2022, doi:10.3390/ani12131601_

Round 1
Reviewer 1 Report
This manuscript highlights 6 novel SNP identified in a native Chinese breed of cattle for the GHRHR gene and their association with body conformation traits. The manuscript is put together fairly well, English grammar and structure is well done and the rationale is good. It is not a novel topic per se, but does provide novel information about the native breed not previously understood. Below are items of major concern that need to be addressed:
1) Authors spend time in both the simple summary and abstract identifying "correlation" among genotypes and conformation traits. Correlation is not an appropriate word considering that a formal linear model was used to understand association of those genetic markers with conformation traits. Using "correlation" is misleading and should be corrected.
2) The authors spend considerable space and energy on sequence alignment, phylogenetic analysis, motif analysis, and predicted structures for data that was publicly available and not related to the cattle used in this study. It is not clear from the introduction that this type of knowledge adds to published literature and based on databases used, is likely already published in other forms. Why spend so much time on this? It does not add or take away from the other aspects of the manuscript to include it or take it out.
3) For the cattle used, were they all of the same sex? A steer (castrated male) will grow differently than an intact male. Both will also grow differently than a female. This is never specified in the manuscript. This is also relevant when it comes to modeling genotypes relative to conformation traits. Age (in months or days) and sex should be other effects in the model with each SNP genotype or diplotype. Otherwise, you cannot clearly identify if genotype/diplotype is important or not.
4) The authors present haplotype and diplotype results, but do not describe how that was accomplished in the Materials & Methods section. The authors also use "genotype" in reference to a pair of haplotypes, but that is not accurate wording. It is typical to use "diplotype" so that genotypes of a given marker are distinguished from a pair of haplotypes.
5) Authors mention a couple of times that all the SNP have "three genotypes". It is more precise to say the SNP are all biallelic, which means they can only have 3 genotypes. A genetic marker can have 3 genotypes present in a population but have more than 2 alleles. SNP by nature, however, are typically only biallelic anyway.
6) All of these issues impact the discussion section, so it would be anticipated that the Discussion section is updated significantly following these changes. The authors should also try to improve readability of some tables due to cells within a row having text run over.
Reviewer 2 Report
Zhao et al identified six SNPs in GHRHR which might be important for body conformation traits. Generally, the approach is reasonable. I have some minor suggestions.
Line 11, 13, and 18: Write the full names of SNPs and GHRHR
Line 15: Should change growth to body conformation traits as the authors did not work on growth traits here.
Line 21: Correlation or association? Which methods are used for analyses, should be mentioned
here?
Line 22: Remove “The results showed that”
Line 22-23: six SNPs were identified in the GHRHR gene, and all SNPs have three genotypes: might change to “six polymorphic SNPs were identified in the GHRHR gene”.
Line 24: “closely related” should change to significantly correlated? Why did the authors perform the correlation analyses?
Line 22-26: The authors did many other analyses regarding Species homology, phylogenetic tree, and conservative estimation, why did not mention them here.
Line 96-101: What did not the authors measure the bodyweight of animals.
Line 104-107: What did the authors choose their primers, and which regions of the genes were amplified.
Line 117-119: which primers were used for sequencing, did the authors purify the PCR project before sequencing?
Line 131: Remove International Business Machines,
Line 137: Change averaged values to mean
Line 138: How many sires do the authors have for the resource population?
The age should use as a covariate in the model instead of a fixed effect.
Line 176-178: The information of intron/exon might add to table 1 together with primer information.
Tables 4 and 5 should move to the supplementary files, they are not so important for the manuscript. The haplotype analyses based on only 6 SNPs do not have any potential use. I do not recommend it for the current manuscript.
Tables 6 and 7 need to reformat and keep a and b in superscript.
Line 268: single SNP remove the single
Line 379: Change combined genotypes to haplotypes. Also, in the able change genotypes to haplotypes. The gene names in the title should be in Italics.
Line 348: remove MAS, it is not necessary
Reviewer 3 Report
I have no comments for that manuscript
Author Response
Thank you very much!
Round 2
Reviewer 1 Report
The manuscript has clearly been improved and the authors addressed almost all of the concerns originally noted. Two items that should still be addressed:
1) The introduction should have some rationale statement indicating why the sequence alignment, phylogenetic analysis, motif analysis, and predicted structures are being investigated. The authors response as to why is appropriate, but that is still not clearly conveyed in the introduction. The authors do provide further background on other species, so the primary issue is a sentence or two summarizing the lack of published comparisons (homology) across these species. I do still wonder if this is not already available through NCBI or similar database though.
2) I think the modeling has been improved and the authors response again makes sense. If the final model followed a previous study from the group, I do think it is beneficial to note that either in the methods or results section to indicate agreement. I could not find a statement to that effect very clearly though.
Some additional minor thoughts that I came across during this read include:
Introduction:
Line 30: Do you mean “diplotype” instead of “genotype”?
Line 41: “Bos taurus” and “Bos indicus” should be in italics
Line 42-43: What do the authors means by “specialized” exactly? What does “excellent quality” refer to? This could be a number of traits in cattle. Improve precise wording here.
Line 59: Should “XinongSannen” have a space – sounds like a line of the Sannen goats called “Xinong”
Lastly, the extensive edits (which are appreciated) did create a lot of grammatical errors. I encourage the authors and editorial staff to review and fix those as they were too numerous to list here. Some of those errors are lack of precise or concise wording. For example:
Line 11: “play a good effect on” is imprecise wording – suggest “… can impact the economic…” or “… can influence the economic…”
